# An Evaluation of the Impact of a Multicomponent Stop Smoking Intervention in an Irish Prison

**DOI:** 10.3390/ijerph182211981

**Published:** 2021-11-15

**Authors:** Andrea Bowe, Louise Marron, John Devlin, Paul Kavanagh

**Affiliations:** 1Health Intelligence Unit, Strategic Planning and Transformation, Health Service Executive, Dublin, Ireland; paul.kavanagh@hse.ie; 2Department of Public Health, Health Service Executive, Dr. Steevens’ Hospital, Dublin, Ireland; louise.marron@hse.ie; 3Irish Prison Service Irish, IDA Business Park, Ballinalee Road, Longford, Ireland; jbdevlin@irishprisons.ie; 4Department of Public Health and Epidemiology, Royal College of Surgeons in Ireland, Dublin 2, Ireland

**Keywords:** tobacco control, prisoner health, multi-component intervention

## Abstract

The disproportionately high prevalence of tobacco use among prisoners remains an important public health issue. While Ireland has well-established legislative bans on smoking in public places, these do not apply in prisons. This study evaluates a multi-component tobacco control intervention in a medium security prison for adult males in Ireland. A stop-smoking intervention, targeting staff and prisoners, was designed, implemented, and evaluated with a before-and-after study. Analysis was conducted using McNemar’s test for paired binary data, Wilcoxon signed rank test for ordinal data, and paired T-tests for continuous normal data. Pre-intervention, 44.3% (*n* = 58) of the study population were current smokers, consisting of 60.7% of prisoners (*n* = 51) and 15.9% of staff (*n* = 7). Post-intervention, 45.1% of prisoners (*n* = 23/51) and 100% of staff (*n* = 7/7) who identified as current smokers pre-intervention reported abstinence from smoking. Among non-smokers, the proportion reporting being exposed to someone else’s cigarette smoke while being a resident or working in the unit decreased from 69.4% (*n* = 50/72) pre-intervention to 27.8% (*n* = 20/72) post-intervention (*p* < 0.001). This multicomponent intervention resulted in high abstinence rates, had high acceptability among both staff and prisoners, and was associated with wider health benefits across the prison setting.

## 1. Introduction

Internationally, there is a high prevalence of smoking among those in prison, which exceeds community prevalence rates by 1.7–8.0-fold [1]. In Ireland, the prevalence of smoking among prisoners has been estimated at 86.0%, approximately 3.7 times the population prevalence [2]. The disproportionately high prevalence of smoking among the prisoner population is multifactorial and likely reflects the higher prevalence of smoking among deprived and socioeconomically disadvantaged populations [3]. It may also reflect the fact that smoking is a part of prison culture [4], and it may be a mechanism of coping with stress, anxiety, or boredom [5] or even a currency that can be exchanged for goods or services within the prison environment [5]. However, the high prevalence of smoking may also reflect under-met health needs in a vulnerable population and the absence of targeted prevention and cessation initiatives within the prison setting to support quitting [6]. The long-term effects of incarceration on physical and mental health are well-documented and include psychological and social effects, as well as effects on cognitive function [7]. Smoking has long-term consequences for health and is a modifiable risk factor which could improve the overall health of prisoners during and after incarceration.

On 29 March 2004, Ireland became the first country in the world to implement legislation banning smoking in the workplace, including in bars and restaurants [8,9,10,11]. However, prisons are defined as a place where the prohibition of tobacco on a premises does not apply [10].

Prisons are a complex setting in terms of tobacco control policy and legislation. For staff, the prison setting is a workplace. Exposure to second-hand smoke (SHS) increases the risk of lung cancer, respiratory illness, and cardiovascular disease and therefore threatens a safe and healthy working environment [12,13]. For the prisoner population, the setting is a home. For those prisoners who do not smoke, exposure to SHS also threatens to impede upon civil rights [4], while for prisoners who smoke, introducing a complete ban may further strip away the few remaining individual rights and privileges. Interventions to tackle tobacco use and second-hand smoke exposure can be challenging to implement, with concerns voiced that limiting access to tobacco products can lead to increased violence in prison setting, albeit evidence to validate this concern is mixed, and there are reports of reductions in violence following implementation [14,15,16].

In the absence of supportive smoking cessation interventions, implementation of complete institutional smoking bans in the prison setting has not been shown to be effective in changing long-term smoking behaviours among prisoners [17,18]. However, legislative and institutional smoking bans are powerful tobacco control interventions that have been proven to reduce exposure to second-hand smoke (SHS), leading to improvements in morbidity and mortality from smoking-related illnesses [19,20].

While some countries have implemented full smoking bans in prison settings and demonstrated reductions in SHS exposure, [21] partial ban policies are used in other countries, including Ireland [22,23,24]. Prisoners in Ireland are not entitled to smoke in prison but may do so with the permission of the prison governor. Smoking is confined to the prisoner’s cell or to outdoor areas [22]. While partial bans do not eliminate SHS exposure, they may reduce it, and they may also be a step toward the creation of smoke-free areas or corridors for those prisoners who do not smoke.

In 2018, the Irish Prison Service (IPS) in collaboration with the Health Service Executive (HSE) Tobacco Free Ireland Programme designed and delivered a multi-component smoking cessation intervention on one site so as to determine how tobacco control might be strengthened across the service within the context of current legislation. This paper describes the evaluation, which aimed to measure the impact of the multi-component smoking cessation intervention on the behaviours, attitudes, self-reported health, and exhaled carbon monoxide (CO) of prisoners and staff in a large Dublin prison.

## 2. Materials and Methods

### 2.1. Setting

The study setting was a closed, medium security prison for adult males in Dublin, Ireland, which is one of the largest prisons in the country [23]. The intervention took place in the “Progression Unit”, which consists of three divisions and in total can accommodate 171 prisoners. Prisoners are carefully selected for the Progression Unit according to pre-defined criteria, which include being drug free, being free of disciplinary sanctions, and signing a contract agreeing to the rules and regulations of the unit. All prisoners in the unit follow a structured rehabilitation and training timetable. Those not engaging with services or activities are relocated back to the main prison. The smoking policy as set out by the IPS restricts smoking to certain areas in the prison, which usually include the prisoner’s cell and designated outdoor areas. Prisoners do not have a right to smoke but may do so with the permission of the prison Governor [22,24].

### 2.2. Population

The target population for this study was prisoners and staff in the Progression Unit. The prisoner population is a transient population; prisoners may be transferred out of the unit or may leave prison due to sentence completion. All prisoners and staff present at the time of the pre-and post-intervention survey were invited to participate. A flow chart describing the study population is contained in Appendix A. For the purposes of this evaluation, the study population comprised of the longitudinal cohort of prisoners (*n* = 87) and staff (*n* = 48) who completed both the pre-intervention and post-intervention questionnaire and who were present in the progression unit during the implementation of the intervention.

Prisoners were invited to participate verbally by prisoner advocates (prisoners who volunteer with the Irish Red Cross). Both staff and prisoners received letters of invitation to the study. Participation was voluntary, and careful planning, including involvement of the Irish Red Cross, ensured that any potential for tacit coercion was mitigated.

### 2.3. Intervention

The multicomponent intervention focussed on three domains:(1)Leadership (including peer leadership) and governance through the establishment of a multidisciplinary intervention team, including the prison governor and trained smoking cessation staff;(2)Building personal skills through education programmes (including peer-led education), provision of a behavioural support programme for smoking cessation, including scheduled weekly support groups, unscheduled peer support and the provision of nicotine replacement therapy (NRT);(3)Creating supportive environments through the establishing of a smoke-free landing in the progression unit, refurbishing this area, including the fit-out of a recreation area, and a “Quitters Day” to celebrate prisoners who successfully quit.

### 2.4. Data Collection

In September 2019, a voluntary pre-intervention survey was conducted among staff and prisoners. The questionnaires for the prisoner population were interviewer-administered by members of the intervention team. The questionnaires for staff were self-administered. In February and March 2020, four months after the intervention was introduced, a post-intervention survey was administered in the same way.

### 2.5. Measures

Smoking status pre-intervention, abstinence rates post-intervention, nicotine dependency (using the Fagerstrom Test for Nicotine Dependence), self-reported exposure to second-hand smoke and self-rated health were measured using a pre- and post-intervention questionnaire, a modified version of which is contained in Appendix A. For those who consented, exhaled CO was measured at the time of the pre-intervention and post-intervention survey using an exhaled CO monitor (MD Diagnostics Ltd, United Kingdom, CO Check Pro). An exhaled CO level above 10 parts per million (ppm) would be a typical reading for a smoker; a level of 3–9 ppm is also typical for a smoker or someone exposed to SHS; a level between 1–4 ppm would be expected for a non-smoker [25,26].

### 2.6. Statistical Analysis

Participants were assigned a unique anonymous identifier which was used to identify the longitudinal cohort who completed both the pre- and post-intervention questionnaire. Statistical analysis was performed using SPSS v26.0 (IBM, New York, NY, USA). The population was described using counts and percentages for categorical data. Normally distributed continuous data were described using means and standard deviations and non-normal data using medians and interquartile ranges. Paired before and after data were analysed using McNemar’s test for paired binary data, Wilcoxon signed rank test for ordinal data, and paired T-tests for continuous normal data. A *p*-value of <0.05 was deemed statistically significant.

### 2.7. Ethical Considerations

Ethical approval for this study was granted by the Royal College of Physicians of Ireland Research Ethics Committee, and the study was also approved by the IPS.

## 3. Results

### 3.1. Sociodemographic Characteristics

The characteristics of the study population are described in Table 1. Overall, 87.8% (*n* = 115) were male, and the mean age was 41.1 ± 10.1 years. Among prisoners, the mean age was 37.2 ± 10.0 years. For 57.8% (n = 48), this was their first time in prison. Prior to entering prison 55.4% (*n* = 46) were in regular employment. Among staff, 66.7% (*n* = 32) were male, and the mean age was 48.0 ± 5.3 years. The majority were employed as prison officers, and the mean duration of time working in the progression unit was 13.5 ± 7.6 years.

### 3.2. Smoking Behaviours Pre-Intervention

Overall, 44.3% (*n* = 58) of study population were current smokers. The prevalence of smoking was 60.7% (*n* = 51) among prisoners and 15.9% (*n* = 7) among staff. Never use was reported by 42.6% of staff (*n* = 20) and 22.6% of prisoners (*n* = 19).

The mean number of cigarettes smoked daily was 12.5 ± 7.5 for prisoners and 12.3 ± 8.4 for staff. For prisoners who smoked, 100% (*n* = 51) reported using hand rolled cigarettes. All staff who smoked reported using manufactured cigarettes.

The characteristics of all those who completed the pre-intervention questionnaire are described in Appendix A.

### 3.3. Abstinence from Smoking Post-Intervention

The post-intervention abstinence rates, measured four months after the intervention, are described in Table 2. Overall, post-intervention, 22.0% (*n* = 29) of study population reported being current smokers (*n* = 28 who remained current smokers and *n* = 1 who identified as a past smoker pre-intervention but current smoker post-intervention). Two abstinence rates were calculated with regard to standard procedures, one based on the total baseline population who completed the pre-intervention survey and the other based on the longitudinal cohort who completed pre- and post- intervention surveys [25,26]. Of the 58 participants who identified as smokers prior to the intervention, 51.7% (*n* = 30) reported being abstinent from smoking on the post-intervention questionnaire. Based on the denominator of *n* = 108 smokers in the total baseline population, and assuming that those who did not complete the post-intervention questionnaire continued to smoke, the abstinence rate post-intervention was 27.8%.

Among the 51 prisoners who were smokers pre-intervention, *n* = 23 (45.1%) reported being abstinent post-intervention. Based on the denominator of *n* = 90 smokers in the complete baseline prisoner population (Appendix A, the abstinence rate post-intervention was 25.6%. No staff member identified as a current smoker post-intervention (100% abstinence rate). Based on the denominator of *n*=18 smokers in the complete baseline staff population, the abstinence rate was 38.9%.

The baseline Fagerstrom Nicotine Dependency Categories between prisoners who did and did not quit did not markedly differ (Appendix A). Among the *n* = 23 prisoners who reported abstinence post intervention, the participation and completion rate for the intervention was 100%. This was significantly higher than among those who continued to smoke, for whom the participation rate was 37.5% (*n* = 9) and the completion rate 33.3% (*n* = 8). All successful quitters reported using a form of NRT on their last quit attempt.

### 3.4. Changes in Self-Reported Exposure to Second-Hand Smoke and Self-Rated Health

The changes in self-reported exposure to SHS and self-rated health are described in Table 3. There was a decrease in self-reported SHS exposure post-intervention (69.2% vs. 35.4%, *p* < 0.001). Among non-smokers, the proportion reporting being exposed to someone else’s cigarette smoke while being a resident or working in the unit decreased from 69.4% (*n* = 50/72) pre-intervention to 27.8% (*n* = 20/72) post intervention (*p* < 0.001).

There were significant differences in self-rated health between pre- and post-intervention surveys for quitters and non-smokers. No significant difference was seen for continuing smokers. Among quitters, 50.0% (*n* = 14) reported better self-rated health post-intervention, with 39.2% (*n* = 11) reporting no change and 10.7% (*n* = 3) rating their health worse than before. Among non-smokers, 42.2% (*n* = 30) reported better self-rated health post intervention, 38% (*n* = 27) reported no change, and 19.7% (*n* = 14) reported worse self-rated health.

There were also statistically significant differences in the proportion reporting respiratory and sensory irritation symptoms pre- and post-intervention. Pre-intervention, 43.9% (*n* = 54) of participants reported at least one respiratory symptom and 43.2% (*n* = 54) at least one sensory irritation symptom, compared to 25.2% (*n* = 31) and 28.8% (*n* = 36) respectively, post-intervention. Among non-smokers, the proportion reporting sensory symptoms was lower post-intervention compared with pre-intervention, this difference was significant (41.2% versus 22.1%, *p* = 0.004).

### 3.5. Changes in Exhaled Carbon Monoxide

The pre- and post-intervention exhaled CO levels are described in Table 4. There was a mean decrease in exhaled CO among quitters of 9.63 ± 6.88 ppm, and this was statistically significant (*p* < 0.001). The mean CO post-intervention for quitters was 3.26 ± 1.93 ppm, consistent with that of a non-smoker [23,24]. For non-smokers, there was a mean decrease of 1.00 ± 2.49 ppm post-intervention; these levels were consistent with that of a non-smoker both pre- and post-intervention. Among those who continued smoking, there was no statistically significant difference in the mean number of cigarettes smoked pre- and post-intervention. However, there was a decrease in the mean exhaled CO of 4.93 ± 7.51 ppm post-intervention.

### 3.6. Changes in Attitudes to Smoking

The changes in attitudes to smoking are described in Table 5. More than 93% of both prisoners and staff either agreed or strongly agreed with the introduction of a smoke-free landing within the unit. This was consistent at both pre- and post-intervention surveys.

The majority of prisoners (87.8%, *n* = 72) and staff (56.8%, *n* = 25) either disagreed or strongly disagreed with the statement that “smoking should not be allowed in prisoners cells”; this remained relatively unchanged post-intervention. Post-intervention, there was a statistically significant difference in the proportion of prisoners and staff who agreed that supports were sufficient to encourage them to quit, increasing from 73.2% to 98.8% among prisoners and from 42.2% to 75.6% among staff.

## 4. Discussion

This study evaluated the impact of a multi-component smoking cessation intervention on the behaviours, attitudes, self-reported health, and exhaled CO of prisoners and staff in a Dublin prison. It has confirmed that high smoking prevalence continues to be a key health challenge among prisoners. Pre-intervention smoking prevalence among prisoners was 60.7%, approximately 3.5 times the reported population level prevalence of 17%, emphasising the need for effective tailored interventions in this vulnerable cohort. However, this study demonstrates that it is feasible to effectively meet the tobacco use-related health needs in this vulnerable population group through well-led implementation of a carefully designed multi-component tobacco control intervention. An abstinence rate of 51.7% at four months follow-up was achieved among the longitudinal cohort of prisoners and staff who were present throughout the intervention, with 45.1% among prisoners and 100% among staff. Furthermore, wider benefits included a reduction in self-reported exposure to SHS among both quitters and non-smokers and a reduction in sensory irritation symptoms among non-smokers.

Other studies have previously demonstrated that high smoking prevalence among people in prisons can be positively impacted through smoking cessation interventions [27]. The quit rate achieved compares well with previously published literature examining smoking cessation interventions in prison settings, which have reported quit rates between 12% and 82%, varying with the method of estimation, duration post-intervention, and study population [17,27]. However, the wide range of reported quit rates raises questions around the optimum design and implementation of smoking cessation interventions in prison settings.

Two distinguishing features of the intervention implemented in this study were inclusion of both staff and prisoners in the target population and the multicomponent design. Few published studies of smoking cessation interventions in a prison setting have sought to address the needs of both staff and prisoners. Adopting a whole-unit approach was an important facet of this intervention which enabled a system-wide change and helped to create a supportive environment for those making quit attempts. This aligns with the World Health Organisation’s recommendation that a settings-based approach is taken to promote health in prisons [28].

The multi-component intervention delivered in this study included peer-led education, psychological support in the form of scheduled weekly support groups and unscheduled peer support, provision of NRT, and the creation of a smoke-free landing within the prison. Pharmacological therapy alone or combined with behavioural intervention increases quit rates among the general population; however, multicomponent interventions are increasingly being used for improving lifestyle behaviours, such as smoking, and have been shown to increase long-term abstinence more than usual care or counselling alone [29]. The intervention performed in this study aligns with existing evidence for effective multicomponent interventions which should include raising awareness of the problem, education, motivation, behavioural change, and medications [29]. A pilot study conducted in a maximum-security prison in Australia examined the feasibility and effectiveness of a multi-component smoking cessation intervention involving brief cognitive behavioural therapy, NRT, bupropion, and self-help resources. Eligible participants included those who smoked more than 30 g of tobacco/week, who expressed intention/motivation to quit, and who were in prison for at least a further three months. Prisoners with cardiovascular disease; certain psychiatric disorders; and those receiving psychotropics, antidepressants, or detoxifcation were excluded. At six-months follow up, 26% of prisoners remained continuously abstinent from smoking, and 37% reported abstinence in the previous seven days [30]. The influence of other variables such as age, education, number of prison terms, and illicit drug use on abstinence were also examined, and those who were continuously absent were more likely to be younger and to have used an illicit drug, but this analysis was limited by the small sample size. The findings of this study in an Irish prison setting with a larger study population add to this evidence on the feasibility and effectiveness of multi-component interventions in the prison setting.

For those who reported abstinence post intervention, there were large and significant decreases in exhaled breath CO, verifying the self-reported change in smoking behaviour. Improved self-rated health post-intervention was reported among 50% of those who quit, and the prevalence of self-reported respiratory symptoms (cough, dyspnoea, wheeze, or sputum production) decreased from 67% pre-intervention to 19% post intervention.

However, there were also wider benefits observed in this study beyond those of individuals. Among non-smokers, there were significant decreases in the proportion reporting exposure to SHS and in the proportion reporting sensory irritation symptoms. These benefits were achieved through augmenting the partial smoking ban in the prison with a smoke-free landing but without implementing a complete ban.

Internationally, approaches to reducing SHS exposure in prison settings differ. Some countries have gone further than Ireland and introduced complete smoking bans in prisons [21,31]. A complete ban on smoking offers better protection from SHS exposure for non-smokers than partial bans, even when these are augmented as in this study. However, the attitudinal findings in this study suggest that barriers remain within the Irish prison setting to the introduction of complete bans. While, post-intervention, more than 95% of both prisoners and staff agreed that there should be a smoke free landing within the unit, the majority of both prisoners and staff, pre- and post-intervention, disagreed with the statement that smoking should not be allowed in prison cells. Monitoring of staff and prisoner views in Scotland points to a different experience: during implementation of a comprehensive smoking ban, reported concerns about a complete ban reduced and support increased, albeit views among staff were generally more positive at all time-points than views among prisoners [32]. Regular and open dialogue about implementation of a full smoking ban and provision of stop smoking care were key enablers to building support for change in Scotland [3]. Stop smoking care provided to prisoners in Scotland is similar to that provided in this study. One difference is the availability of e-cigarettes, which are available to prisoners in Scotland [33] but are not recommended or available in prisons in Ireland.

There are limitations to the findings of this study, perhaps the most significant being the generalisability of these findings to the wider prison population. The study population consisted of prisoners and staff in the Progression Unit of this prison. Prisoners selected for this unit must meet defined criteria, which include being drug-free and free of behavioural sanctions and a high security risk, and they must engage actively in programmes within the unit. Therefore, this cohort is likely to represent a highly motivated group within the prison. Their motivation for participating in the intervention, evidenced by the exceptionally high participation rates, may extend beyond that of wishing to abstain from smoking. It may be influenced by the need to engage in programmes within the unit or by impending release from prison. These motivations may not apply to the general prison population.

The observational, uncontrolled before-and-after study design does not allow definitive conclusions to be drawn regarding intervention effectiveness, albeit that the effectiveness of key components such as behavioural stop smoking support and NRT are well-established. To overcome literacy challenges for prisoners, the surveys were interviewer-administered and are therefore subject to interviewer bias and response bias. Self-reported quit rates were however validated by the large and significant decrease in exhaled CO in this cohort. Due to challenges around changing staff patterns during the COVID-19 pandemic it is difficult to accurately calculate response rates among staff. The response rate among prisoners, however, was exceptionally high at both pre- and post-intervention surveys, and of the *n* = 87 prisoners present at pre- and post-intervention surveys, 96.5% were included in this study.

Acknowledging these limitations, this study provides a detailed description of the positive impact of a multicomponent intervention with a strong peer-led element that resulted in a self-reported abstinence rate of 52% at four-months post-intervention, had high acceptability among both staff and prisoners, and was associated with wider health benefits across the prison setting.

## 5. Conclusions

The current policy in Irish prisons allows smoking in defined areas of the prison, usually including the prisoner cells and designated outdoor areas [22,24]. There is a careful balance to be struck in upholding dignity of prisoners and in protecting public health for both staff and prisoners who may be exposed to SHS. Countries and even courts have differed in their approach and opinion on this issue. This present study demonstrates that in the absence of mandatory measures, smoking prevalence in prison settings can be successfully reduced in a voluntary capacity using multicomponent interventions with strong peer support. Should a mandatory complete ban be introduced in prisons in future in Ireland, optimising smoking cessation through building on lessons learned from this intervention will provide a necessary platform for implementation. Engagement with prisoners and prison staff will also be required to enable change. Regardless of further policy, the potential to improve smoking cessation care identified in this study must be seized to better meet the complex health needs of this vulnerable population.

## Figures and Tables

**Table 1 ijerph-18-11981-t001:** Sociodemographic and Smoking Characteristics of Study Population.

	Overall (*n* = 132)		Prisoners (*n* = 84)		Staff (*n* = 48)	
	Valid	*n* (%)	Valid	*n* (%)	Valid	*n* (%)
Male gender	131	115 (87.8)	83	83 (100)	48	32 (66.7)
Age in Years, mean (sd)	127	41.1 (10.1)	82	37.2 (10.0)	45	48.0 (5.3)
Prisoners						
Months in prison, median (IQR)			84	5.4 (4.8)		
First time in prison			83	48 (57.8)		
In regular employment prior to prison			83	46 (55.4)		
Staff						
Role in progression unit					48	
Prison officer						36 (75.0)
Other						22 (25.0)
Years working in unit, mean (sd)					47	13.5 (7.6)
Smoking Status Pre-Intervention	131		84		47	
Current smoker		58 (44.3)		51 (60.7)		7 (14.9)
Past smoker		34 (25.9)		14 (16.7)		20 (42.6)
Never smoker		39 (29.8)		19 (22.6)		20 (42.6)
Cigarettes smoked daily, mean (sd)	58	12.5 (7.6)	51	12.5 (7.5)	7	12.3 (8.4)
Age in years when began, mean (sd)	58	15.3 (5.2)	51	15.1 (5.3)	7	17.4 (4.0)
Fagerstrom Category of Smokers	57		51		6	
Low		12 (21.1)		10 (19.6)		2 (33.3)
Low to moderate		14 (24.6)		12 (23.5)		2 (33.3)
Moderate		24 (42.1)		23 (45.1)		1 (16.7)
High		7 (12.3)		6 (11.8)		1 (16.7)

IQR: Interquartile range, sd: standard deviation.

**Table 2 ijerph-18-11981-t002:** Changes in Smoking Behaviours and Abstinence Rates Post- Intervention.

Population	Pre-Intervention Prevalence of Smoking *n* (%)	Post-Intervention Prevalence of Smoking *n* (%)	*p*-Value	Abstinence rate Post-Intervention *n* (%)
Prisoners	51 (60.7)	29 (34.5) *	<0.001	23 (45.1)
Staff	7 (14.9)	0 (0.0)	0.016	7 (100.0)

* *n* = 29 includes one participant who was a non-smoker at pre-intervention and smoker post intervention.

**Table 3 ijerph-18-11981-t003:** Changes in Self-Reported Exposure to Second-hand Smoke and Self-Rated Health.

		Overall (*n* = 130 ^*^)			Quitters (*n* = 30)				Non-Smokers (*n* = 72)				Continued Smokers (*n* = 28)			
	*n*	Pre	Post	*P ^^^*	*N*	Pre	Post	*P ^^^*	*n*	Pre	Post	*P ^^^*	*n*	Pre	Post	*P ^^^*
Any exposure to SHS	130	90 (69.2)	46 (35.4)	<0.001	29	22 (75.9)	7 (24.1)	<0.001	72	50 (69.4)	20 (27.8)	<0.001	28	18 (64.3)	18 (64.3)	1.000
Duration of SHS exposure per day ^^^	118				26				67				23			
Never/almost never		36 (30.5)	73 (61.9)			5 (19.2)	19 (73.1)			22 (32.8)	47 (70.1)			9 (39.1)	7 (30.4)	
<1 h		26 (22.0)	23 (19.5)			4 (15.4)	6 (23.1)			20 (29.9)	12 (17.9)			1 (4.3)	4 (17.4)	
1–5 h		23 (19.5)	12 (10.2)			8 (30.7)	0 (0.0)			10 (14.9)	6 (9.0)			5 (21.7)	5 (21.7)	
6–10 h		24 (20.3)	10 (8.5)			8 (30.7)	1 (3.8)			8 (11.9)	2 (3.0)			7 (30.4)	7 (30.4)	
≥10 h		9 (7.6)	0 (0.0)	<0.001		1 (3.8)	0 (0.0)	<0.001		7 (10.4)	0 (0.0)	<0.001		1 (4.3)	0 (0.0)	0.949
Self-rated health ^$^	128				28				71				28			
Poor		7 (5.5)	2 (1.6)			3 (12.5)	0			2 (2.8)	1 (1.4)			2 (7.1)	1 (3.6)	
Fair		23 (18.0)	15 (11.7)			10 (35.7)	5 (17.9)			8 (11.3)	6 (8.5)			5 (17.9)	4 (14.3)	
Good		42 (32.8)	42 (32.8)			9 (32.1)	12 (42.9)			17 (23.9)	14 (19.7)			15 (53.6)	16 (57.1)	
Very good		37 (28.9)	36 (28.1)			5 (17.9)	7 (25.0)			27 (38.0)	23 (32.4)			5 (17.9)	6 (21.4)	
Excellent		19 (14.8)	33 (25.8)	<0.001		1 (3.6)	4 (14.3)	0.004		17 (23.9)	27 (38.0)	0.013		1 (3.6)	1 (3.6)	0.361
Any resp sx	123	54 (43.9)	31 (25.2)	<0.001	27	18 (66.7)	5 (18.5)	0.001	68	14 (20.6)	9 (13.62)	0.267	27	22 (81.5)	17 (62.9)	0.180
Any sensory sx	125	54 (43.2)	36 (28.8)	0.010	28	14 (50.0)	8 (28.6)	0.146	68	28 (41.2)	15 (22.1)	0.004	28	12 (42.9)	12 (42.9)	1.000

SHS: Second-hand smoke exposure; Resp sx: Respiratory symptom; Sensory Sx: Sensory symptom; ^^^
*p*-values calculated using McNemar chi-square test for binary paired data or Wilcoxon sign rank test for ordinal paired data.^*^
*n* = 130 excludes one participant who started smoking during intervention and one participant for whom smoking status was not reported ^^^ Overall: 63 negative ranks, 13 positive ranks, 42 ties; Quitters: 19 negative ranks, 0 positive ranks, 7 ties; Non-smokers: 36 negative ranks, 6 positive ranks, 25 ties; Smokers: 7 negative ranks, 7 positive ranks, 9 ties. ^$^ Overall: 22 negative ranks, 53 positive ranks, 53 ties; Quitters: 14 positive ranks, 3 negative, 11 ties; Non-smokers: 30 positive ranks, 14 negative ranks, 27 ties; Smokers: 8 positive ranks, 5 negative ranks, 15 ties.

**Table 4 ijerph-18-11981-t004:** Changes in Pre and Post Exhaled Carbon Monoxide.

	Valid	Pre Exhaled CO ^+^	Post Exhaled CO ^++^	Mean Difference	*p*-Value ^^^
	*n*	Mean (sd)-ppm			
Overall	120	8.39 (8.48)	4.58 (5.46)	3.81 (6.20)	<0.001
Quitters	27	12.89 (6.58)	3.26 (1.93)	9.63 (6.88)	<0.001
Non smokers	65	2.71 (2.11)	1.71 (1.41)	1.00 (2.49)	0.002
Continued smokers ^+^	27	17.81 (8.91)	12.89 (5.81)	4.93 (7.51)	0.002

^+^ Measured Exhaled Carbon Monoxide at pre-intervention survey, ^++^ Measured Exhaled Carbon Monoxide at post-intervention survey, *^^^ p*-value calculated using paired *t*-test.

**Table 5 ijerph-18-11981-t005:** Changes in Attitudes.

	Prisoners (n = 84)			Staff (n = 47)		
	Pre	Post	*P ^^^*	Pre	Post	*P ^^^*
Smoking restrictions in the unit are adequate						
Strongly agree/agree	66 (78.6)	75 (89.3)		26 (59.0)	33 (75.0)	
Disagree/strongly disagree	18 (21.4)	9 (11.7)	0.093	18 (41.0)	11 (25.0)	0.118
Creation of smoke-free zones within the unit is a good thing ^+^						
Strongly agree/agree	52 (62.7)	64 (77.1)		42 (89.4)	43 (91.5)	
Disagree/strongly disagree	31 (37.3)	19 (20.9)	0.059	5 (10.6)	4 (8.5)	1.000
Smoking should not be allowed in prison cells ^^^^						
Strongly agree/agree	10 (12.2)	11 (13.4)		20 (44.4)	19 (43.2)	
Disagree/strongly disagree	72 (87.8)	71 (86.6)	1.000	25 (56.8)	26 (57.7)	1.000
Supports are sufficient to encourage prisoners/staff to quit smoking ^^^^						
Strongly agree/agree	60 (73.2)	81 (98.8)		19 (42.2)	34 (75.6)	
Disagree/strongly disagree	22 (26.8)	1 (1.2)	<0.001	26 (57.8)	11 (24.4)	0.001
There should be a smoke-free landing in the progression unit ^^^^						
Strongly agree/agree	77 (93.9)	79 (96.3)		44 (93.6)	46 (97.9)	
Disagree/Strongly disagree	5 (6.1)	3 (3.7)	0.625	3 (6.3)	1 (2.1)	0.500

Pre: Pre-intervention survey, Post: Post-intervention survey, SHS: Second-hand smoke, ^^^
*p*-value calculated using McNemar’s chi-square test; ^+^
*n* = 83 in prisoner analysis ^^^^
*n* = 82 in prisoner analysis.

## Data Availability

The data used in this study are not currently publicly available as participants did not consent for their data to be shared publicly.

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
