# Peer review of "An Evaluation of the Impact of a Multicomponent Stop Smoking Intervention in an Irish Prison"

_ijerph, 2021, doi:10.3390/ijerph182211981_

Round 1

Reviewer 1 Report

This paper analyze a relevant topic, as the stop smoking intervention in a prison. The methodology is correct in this type of sample. The sample is compossed of prisoner (n=84) and staff (n=47). The study is centered in mediational analysis and not indicate if it is related at the abstinence or at the reduction. For this, it is necessary include a table with means and SD of smoking consumption, at the begining of the intervention and in end of them. Similarly, it is necessary include the porcentaje of person that are abstinent at the end of the intervention. In the discussion and conclussion the authors need include this topic and discuss the abstinence of the intervention.

Author Response

Response in file attached

Reviewer 2 Report

The topic is realy interested for me. Text is useful. I compared your results (in Ireland) with prophilaxis of smoking in the prisons in my country. I thought about limitations for described intervention.

In my country there are ca. 72 000 people in penitentiary institution. Smoking is very popular, and prophylaxis programs of reductions have little, inappreciable result. Your intervention is at 45%, so it cerates some qiestions.

Background

  1. I regret, there is no psychological, theoretical backgoud. About the situation, attitudes, personality and changing of cognitive abilities, during punishment. As far me this raport is adressed to practicioners, and it was skipped.
  2. … prisons are defined as a place where the prohibition of tobacco on a premises does not apply [9]. Prisons are a complex setting in terms of tobacco control policy and legislation. For staff, the prison setting is a workplace. Exposure to second-hand smoke [SHS] increases the risk of lung cancer, respiratory illness, and cardiovascular disease and therefore threatens a safe and healthy working environment [10,11,12]. For the prisoner population, the setting is a home. For those prisoners who do not smoke, exposure to SHS also threatens to impede upon civil rights [4], while for prisoners who smoke, introducing a com- 49 plete ban may further strip away the few remaining individual rights and privileges.

Please describe more details about smoking in different types of prison. You focused on „medium security prison”, „progression unit”, and that’s ok.

But smoking (origins, reasons, effects) must be different in high level security prison. Time, stress, space, relationships differ in many ways, whan a man leaves in high and medium security.

I  stimulate You to reflection about informal aspects of smoking, banning. For instance: stop smoking and aggresiopn, extra prohibitions and revolt? In medium security, in progression unit it isn’t predictable, but in high security, too?

Participants:

Prisoners were invited to participate verbally by prisoner advocates [prisoners who volunteer with the Irish Red Cross].  (limitations) This may affect their motivations for behaviour change and may limit the generalisability of these findings to wider prison populations

Do You predict any external motivation of prisoners? For instance- it is goog to take part, „show them” my determiation on pogression, because it maight be  profitable in future? Have You any reflection/a piece of information about motivation of participants?

Prison Staff- 67% of participants were ca. 50 years old. Do You know sth about their motivation? What about motivation of younger participants?  

Limitations for intervention (as far me very sweeping.)

The point is- You selected carefully participants.

All of them there are in „progression”. Consequently results of intervention are impress. But thres is only a little part of prison’s life. There are a lot of limitations.

 As a reader I thought, Your interwention and results are disruptive (Abstract- Post 19 intervention, the self-reported quit rate was 45.1% among prisoners and 100% among staff, verified  with reductions in exhaled CO; discussion Other studies have previously demonstrated that high smoking prevalence among people in prisons can be positively impacted through smoking cessation interventions.), but realy is it?

This may affect their motivations for behaviour change and may limit the generalisability of these findings to wider prison populations

Do You predict any implied variables-  For instnace: „Building personal skills through education programmes”- is predictable the level od IQ, cognitive abilities were higher in that selected gropu? Are You certain it isn’t the result of your interwention, but cognitive abilities in the selected group?

Discusion

A study conducted in a maximum-security prison in Australia [n=30] examined the feasibility and effectiveness of a multi-component smoking cessation intervention involving brief cognitive behav ioural therapy- what about participants in Australia? Were there selected in „progressive unit” or not? Have You more details making an impact on results and their interpretation? Were in Australia addictional variables?

Thank You for fruitful article.

Author Response

Responses in attached file.

Round 2

Reviewer 1 Report

Accept the paper.